# Once-for-All Adversarial Training: In-Situ Tradeoff between Robustness and Accuracy for Free

**Haotao Wang**[†,*] **Tianlong Chen**[†,*] **Shupeng Gui**[◇] **, Ting-Kuei Hu**[¶],
**Ji Liu**[‡] and **Zhangyang Wang**[†]

[†]Department of Electrical and Computer Engineering, The University of Texas at Austin
[◇]Department of Computer Science, University of Rochester
[¶]Department of Computer Science and Engineering, Texas A&M University
[‡]Ytech Seattle AI lab, FeDA lab, AI platform, Kwai Inc
[†]*{htwang, tianlong.chen, atlaswang}@utexas.edu*
[◇]*sgui.aca@gmail.com*    [¶]*tkhu@tamu.edu*    [‡] *ji.liu.uwisc@gmail.com*

## Abstract

Adversarial training and its many variants substantially improve deep network robustness, yet at the cost of compromising standard accuracy. Moreover, the training process is heavy and hence it becomes impractical to thoroughly explore the trade-off between accuracy and robustness. This paper asks this new question: *how to quickly calibrate a trained model in-situ, to examine the achievable trade-offs between its standard and robust accuracies, without (re-)training it many times?* Our proposed framework, *Once-for-all Adversarial Training* (**OAT**), is built on an innovative model-conditional training framework, with a controlling hyper-parameter as the input. The trained model could be adjusted among different standard and robust accuracies "for free" at testing time. As an important knob, we exploit dual batch normalization to separate standard and adversarial feature statistics, so that they can be learned in one model without degrading performance. We further extend OAT to a *Once-for-all Adversarial Training and Slimming* (**OATS**) framework, that allows for the joint trade-off among accuracy, robustness and runtime efficiency. Experiments show that, without any re-training nor ensembling, OAT/OATS achieve similar or even superior performance compared to dedicatedly trained models at various configurations. Our codes and pretrained models are available at: https://github.com/VITA-Group/Once-for-All-Adversarial-Training.

## 1   Motivation and background

Deep neural networks (DNNs) are nowadays well-known to be vulnerable to adversarial examples [1, 2]. With the growing usage of DNNs on security sensitive applications, such as self-driving [3] and bio-metrics [4], a critical concern has been raised to carefully examine the worst-case accuracy of deployed DNNs on crafted attacks (denoted as *robust accuracy*, or *robustness* for short, following [5]), in addition to their average accuracy on standard inputs (denoted as *standard accuracy*, or *accuracy* for short). Among a variety of adversarial defense methods proposed to enhance DNN robustness, adversarial training (AT) based methods [5, 6, 7] are consistently top-performers.

While adversarial defense methods are gaining increasing attention and popularity in safety/security-critical applications, their downsides are also noteworthy. Firstly, most adversarial defense methods, including adversarial training, come at the price of compromising the standard accuracy [8]. That

inherent *accuracy-robustness trade-off* is established both theoretically, and observed experimentally, by many works [5, 9, 10, 11, 12, 13]. Practically, most defense methods determine their accuracy-robustness trade-off by some pre-chosen hyper-parameter. Taking adversarial training for example, the training objective is often a weighted summation of a standard classification loss and a robustness loss, where the trade-off coefficient is typically set to an empirical value by default. Different models are then trained under this same setting to compare their achievable standard and robust accuracies.

However, in a practical machine learning system, the requirements of standard and robust accuracies may each have their specified bars, that are often not naturally met by the "default" settings. While a standard trained model might have unacceptable robust accuracy (*i.e.*, poor in "worst case"), an adversarially trained model might compromise the standard accuracy too much (*i.e.*, not good enough in "average case"). Moreover, such standard/robust accuracy requirements can vary *in-situ* over contexts and time. For example, an autonomous agent might choose to perceive and behave more cautiously, *i.e.*, to prioritize improving its robustness, when it is placed in less confident or adverse environments. Besides, more evaluation points across the full spectrum of accuracy-robustness trade-off would also provide us with a more comprehensive view of the model behavior, rather than just two evaluation points (*i.e.*, standard trained and adversarially trained with default settings).

Therefore, practitioners look for convenient means to explore and flexibly calibrate the accuracy-robustness trade-off. Unfortunately, most defense methods are intensive to train, making it tedious or infeasible to naively train many different configurations and then pick the best. For instance, adversarial training is notoriously time-consuming [14], despite a few acceleration strategies (yet with some performance degradation) being proposed recently [14, 15].

## 1.1 Our contributions

Motivated by the above, the core problem we raise and address in this paper is:

*How to quickly calibrate a trained model in-situ, to examine the achievable trade-offs between its standard and robust accuracies, without (re-)training it many times?*

We present a novel *Once-for-all Adversarial Training* (**OAT**) framework that achieves this goal for the first time. OAT is established on top of adversarial training, yet augmenting it with a new *model-conditional training* approach. OAT views the weight hyper-parameter of the robust loss term as a user-specified input to the model. During training, it samples not only data points, but also *model instances* from the objective family, parameterized by different loss term weights. As a result, the model learns to condition its behavior and output on this specified hyper-parameter. Therefore at testing time, we could adjust between different standard and robust accuraices "for free" in the same model, by simply switching the hyper-parameters as inputs.

When developing OAT, one technical obstacle we discover is that the resultant model would not achieve the same high standard/robust accuracy, as it could achieve when being training dedicatedly at a specific loss weight. We investigate the problem and find it to be caused by the unaligned and somehow conflicting statistics between standard and adversarially learned features [16]. That makes it challenging for OAT to implicitly pack different standard/adversarially trained models into one. In view of that, we customize a latest tool called *dual batch normalization*, originally proposed by [17] for improving (standard) image recognition, to separate the standard and adversarial feature statistics in training. It shows to become an important building block for the success of OAT.

A further extension we studied for OAT is to integrate it to the recently rising field of *robust model compression or acceleration* [18, 19, 20], for more flexible and controllable deployment purpose. We augment the model-conditional training to further condition on another *model compactness* hyper-parameter, implemented as the channel width factor in [21]. We call this augmented framework *Once-for-all Adversarial Training and Slimming* (**OATS**). OATS leads to trained models that can achieve in-situ trade-off among accuracy, robustness and model complexity altogether, by adjusting the two hyper-parameter inputs (robustness weight and width factor). It is thus desired by many resource-constrained, yet high-stakes applications.

Our contributions can be briefly summarized as below:

- A novel OAT framework that addresses a new and important goal: in-situ "free" trade-off between robustness and accuracy at testing time. In particular, we demonstrate the importance of separating standard and adversarial feature statistics, when trying to pack their learning in one model.

- An extension from OAT to OATS, that enables a joint in-situ trade-off among robustness, accuracy, and the computational budget.

- Experimental results show that OAT/OATS achieve similar or even superior performance, when compared to dedicatedly trained models. Our approaches meanwhile cost only one model and no re-training. In other words, they are **free but no worse**. Ablations and visualizations are provided for more insights.

## 2  Preliminaries

Given the data distribution $\mathcal{D}$ over images $x \in \mathbb{R}^d$ and their labels $y \in \mathbb{R}^c$, a standard classifier (*e.g.*, a DNN) $f : \mathbb{R}^d \to \mathbb{R}^c$ with parameter $\theta$ maps an input image to classification probabilities, learned by empirical risk minimization (ERM):

$$\min_\theta \mathbb{E}_{(x,y)\sim\mathcal{D}} \ \mathcal{L}(f(x;\theta), y)$$

where $\mathcal{L}(\cdot, \cdot)$ is the cross-entropy loss by default: $\mathcal{L}(f(x;\theta), y) = -y^T \log(f(x;\theta))$.

**Adversarial training**   Numerous methods have been proposed to enhance DNN adversarial robustness, among which adversarial training (AT) based methods [6] are arguably some of the most successful ones. Most state-of-the-art AT algorithms [5, 6, 22, 23, 24] optimize a hybrid loss consisting of a standard classification loss and a robustness loss term:

$$\min_\theta \mathbb{E}_{(x,y)\sim\mathcal{D}} \left[ (1-\lambda)\mathcal{L}_c + \lambda\mathcal{L}_a \right] \tag{1}$$

where $\mathcal{L}_c$ denotes the classification loss over standard (or clean) images, while $\mathcal{L}_a$ is the loss encouraging the robustness against adversarial examples. $\lambda$ is a fixed weight hyper-parameter. For example, in a popular form of AT called PGD-AT [6] and its variants [8]:

$$\mathcal{L}_c = \mathcal{L}(f(x;\theta), y), \mathcal{L}_a = \max_{\delta \in \mathcal{B}(\epsilon)} \mathcal{L}(f(x+\delta;\theta), y) \tag{2}$$

where $\mathcal{B}(\epsilon) = \{\delta \mid \|\delta\|_\infty \le \epsilon\}$ is the allowed perturbation set. TRADES [5] use the same $\mathcal{L}_c$ as PGD-AT, but replace $\mathcal{L}_a$ from cross-entropy to a soft logits-pairing term. In MMA training [24], $\mathcal{L}_a$ is to maximize the margins between correctly classified images.

For all these state-of-the-art AT-type methods, the weight $\lambda$ has to be set a fixed value before training, to determine the relative importance between standard and robust accuracies. Indeed, as many works have revealed [8, 5, 9], there seems to exist a potential trade-off between the standard and robust accuracies that a model can achieve. For example, in PGD-AT, $\lambda = 1$ is the default setting. As a result, the trained models will only demonstrate one specific combination of the two accuracies. If one wishes to trade some robustness for more accuracy gains (or vice versa), there seems to be no better option than re-training with another $\lambda$ from scratch.

**Conditional learning and inference**   Several fields have explored the idea to condition a trained model's inference on each testing input, for more controllablity. Examples can be firstly found from the dynamic inference literature, whose main idea is to dynamically adjust the computational path per input. For example, [25, 26, 27] augment DNN with multiple side branches, through which early predictions could be routed. [28, 29] allowed for an input to choose between passing through or skipping each layer. One relevant prior work to ours, the slimmable network [21], aims at training a DNN with adjustable channel width during runtime. The authors replaced batch normalization (BN) by switchable BN (S-BN), which employs independent BNs for each different-width subnetwork. The new network is then trained by optimizing the average loss across all different-width subnetworks. Besides, many generative models [30, 31] are able to synthesize outputs conditioned on the input class labels. Other fields where conditional inference is popular include visual question answering [32], visual reasoning [33], and style transfer [34, 35, 36, 37]. All those methods take advantage of certain side transformation, by transforming them to modulate some intermediate DNN features. For example, the feature-wise linear modulation (FiLM) [33] layers influence the network computation via a feature-wise affine transformation based on conditioning information.

# 3  Technical approach

## 3.1  Main idea: model-conditional training by joint model-data sampling

The goal of OAT is to learn a distribution of DNNs $f(\cdot, \lambda; \theta) \sim F$, conditioned on $\lambda \sim P_\lambda$, so that different DNNs sampled from this learned distribution, while sharing the set of parameters $\theta$, could have different accuracy-robustness trade-offs depending on the input $\lambda$.

While standard DNN training samples data, OAT proposes to also sample one $f(\cdot, \lambda; \theta) \sim F$ per data. Each time, we set $f$ to be conditioned on $\lambda$: concretely, it will take a hyperparameter $\lambda \sim P_\lambda$ as the input, while using this same $\lambda$ to modulate the current AT loss function:

$$\mathcal{L}(x, y, \lambda) = \mathbb{E}_{(x,y) \sim \mathcal{D}, \lambda \in P_\lambda} \left[ (1 - \lambda) \mathcal{L}_c + \lambda \mathcal{L}_a \right] \tag{3}$$

The gradient w.r.t. each $(x, y)$ is generated from the loss parameterized by the current $\lambda$. Therefore, OAT essentially optimizes a dynamic loss function, with $\lambda$ varying per sample, and forces weight sharing across iterations. We call it *model-conditional training*. Without loss of generality, we use cross-entropy loss for $\mathcal{L}_c$ and $\mathcal{L}_a$ as PGD-AT did in Eq (2): $\mathcal{L}_c = \mathcal{L}(f(x, \lambda; \theta), y), \mathcal{L}_a = \max_{\delta \in \mathcal{B}(\epsilon)} \mathcal{L}(f(x + \delta, \lambda; \theta), y)$. Notice that OAT is also compatible with other forms of $\mathcal{L}_c$ and $\mathcal{L}_a$, such as those in TRADES and MMA training. The OAT algorithm is outlined in Algo. 1.

Our model sampling idea is inspired by and may be intimately linked with *dropout* [38], although the two serve completely different purposes. Applying dropout to a DNN amounts to sampling a sub-network from the original one at each iteration, that consists of all units surviving dropout. So training a DNN with dropout can be seen as training a collection of explosively many sub-networks, with extensive weight sharing, where each thinned network gets sparsely (or rarely) trained. The trained DNN then behaves approximately like an ensemble of those subnetworks, enhancing its generalization. Similarly, OAT also samples a model per-data: yet each sampled model is parameterized by a different loss function, rather than a different architecture. Further, intuitively we could also interpret the OAT

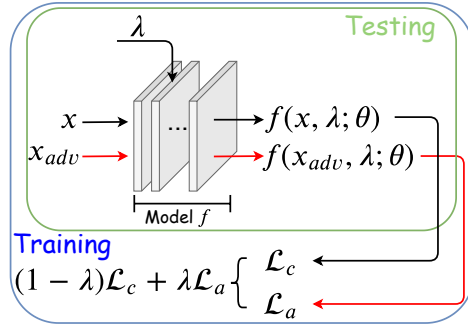

Figure 1: The overall OAT framework illustration. The hyperparameter $\lambda$ is set as both the input and the loss function hyperparameter. It is varied in training and can be specified during testing.

trained model as an ensemble of those sampled models; however, the model output at run time is conditioned on the specified $\lambda$ input which could be considered by "model selection", in contrast to the static "model averaging" interpretation in dropout.

**How to encode and condition on $\lambda$**  We first embed $\lambda$ from a scalar to a high dimensional vector in $\mathbb{R}^d$. While one-hot label is a naive option, we use (nearly)-orthogonal random vectors to encode different $\lambda$s, for better scalability and empirically better performance.

We adopt the module of Feature-wise Linear Modulation (FiLM) [33] to implement our conditioning of $f$ on $\lambda$. Suppose a layer's output feature map is $\boldsymbol{h} \in \mathbb{R}^{C \times H \times W}$, FiLM performs a channel-wise affine transformation on $\boldsymbol{h}$, with parameters $\boldsymbol{\gamma} \in \mathbb{R}^C$ and $\boldsymbol{\beta} \in \mathbb{R}^C$ dependent on the input $\lambda$:

$$FiLM(h_c; \boldsymbol{\gamma}_c, \boldsymbol{\beta}_c) = \boldsymbol{\gamma}_c h_c + \boldsymbol{\beta}_c, \ \boldsymbol{\gamma} = g_1(\lambda), \ \boldsymbol{\beta} = g_2(\lambda)$$

where the subscripts refer to the $c^{th}$ feature map of $\boldsymbol{h}$ and the $c^{th}$ element of $\boldsymbol{\gamma}, \boldsymbol{\beta}$. $g_1$ and $g_2$ are two multi-layer perceptrons (MLPs) with Leaky ReLU. The output for FiLM is then passed on to the next layers in the network. In practice, we perform that affine transformation after every batch normalization (BN) layer, and each MLP has two $C$-dimensional layers.

## 3.2  Overcoming a unique bottleneck: standard and adversarial feature statistics

After implementing the preliminary OAT framework in Section 3.1, an intriguing observation was found: while varying $\lambda$ can get different standard and robust accuracies, those achieved numbers are significantly degraded compared to those from dedicatedly trained models with fixed $\lambda$. Further investigation reveals the "conflict" arising from packing different standard and adversarial features in one model seems to cause this bottleneck, and a split of BN could efficiently fix it.

To illustrate this conflict, we train ResNet34 using PGD-AT, with $\lambda$ in the objective (1) varying from 0 (standard training) to 1 (common setting for PGD-AT), on the CIFAR-10 dataset. We then visualize the statistics of the last BN layer, *i.e.*, the running mean and running variance, as shown in Fig. 2. While smoothly changing $\lambda$ leads to continuous feature statistics transitions/shifts as expected, we observe the feature statistics of $\lambda = 0$ (black dots) seems to be particularly isolated from those of other nonzero $\lambda$s; the gap is large even just between $\lambda = 0$ and 0.1. Meanwhile, all nonzero $\lambda$s lead to largely overlapped feature statistics. That is also aligned with our experiments showing that the preliminary OAT tend to fit either only the cleans images

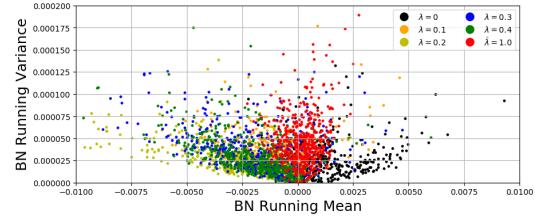

Figure 2: Running mean ($x$-axix) and variance ($y$-axix) of the last BN layer from ResNet34 trained with PGD-AT and fixed $\lambda$ values on CIFAR10. Each model trained with a different $\lambda$ corresponds to 512 elements (the BN mean/variance tensors), that is colored differently (see the legend).

(resulting in degraded robust accuracy) or only adversarial images (resulting in degraded standard accuracy). Therefore, we conjecture that the difference between standard ($\lambda = 0$) and adversarial ($\lambda \neq 0$) feature statistics might account for the challenge when we try to pack their learning together.

We notice a line of very recent works [17, 39]: despite the problem setting and goal being very different with ours, the authors also reported similar difference between standard and adversarial feature statistics. They crafted *dual batch normalization* (dual BN) as a solution, by separating the standard and adversarial feature statistics in two dedicated BNs. That was shown to improve the standard accuracy for image recognition, while adversarial examples act as a special data augmentation in their application setting.

We hereby customize dual BN for our setting. We replace all BN layers with dual BN layers in the network. A dual BN consists of two independent BNs, $BN_c$ and $BN_a$, accounting for the standard and adversarial features, respectively. A switch layer follows to select one of the two BNs to be activated for the current sample. In [17, 39], the authors only handle clean examples (*i.e.*, $\lambda = 0$) as well as adversarial examples generated by "common" PGD-AT, *i.e.*, $\lambda = 1$. Also, they only aim at improving standard accuracy at testing time. The switching policy is thus

---

**Algorithm 1:** OAT Algorithm Outline

**Input:** Training set $\mathcal{D}$, model $f$, $P_\lambda$, maximal steps $T$.
**Output:** Model parameter $\theta$.
**for** $t = 1$ **to** $T$ **do**
    Sample a minibatch of $(\boldsymbol{x}, \boldsymbol{y})$ from $\mathcal{D}$;
    Sample a minibatch of $\lambda$ from $P_\lambda$;
    Generate $\boldsymbol{x}_{adv}$ (using PGD);
    Update network parameter $\theta$ by Eq. (3)
**end for**

---

straightforward for their training and testing. However, OAT needs to take a variety of adversarial examples generated with all $\lambda$s between [0,1], for both training and testing. Based on our observations from Fig. 2, at *both* training and testing time, we route $\lambda = 0$ cases through $BN_c$, while all $\lambda \neq 0$ cases go to $BN_a$. At testing time, the value of $\lambda$ is specified by the user, per his/her requirement or preference between the standard and robust accuracies: a larger $\lambda$ emphasizes robustness more, while a smaller $\lambda$ for better accuracy. As demonstrated in our experiments, such **modified dual BN** is an important contributor to the success of OAT.

### 3.3 Extending from OAT to OATS: joint trade-off with model efficiency

The increasing popularity of deploying DNNs into resource-constrained devices, such as mobile phones, IoT cameras and outdoor robots, have raised higher demands for not only DNNs' performance but also their efficiency. Recent works [18, 19] have emerged to compress models or reduce inference latency, while minimally affecting their accuracy and robustness, pointing to the appealing goal of "co-designing" a model to be accurate, trustworthy and resource-friendly. However, adding the new efficiency dimension further complicates the design space, and makes it even harder to manually examine the possible combinations of model capacities and training strategies.

We extend the OAT methodology to a new framework called *Once-for-all Adversarial Training and Slimming* (**OATS**). The main idea is to augment the model-conditional training to further condition on another *model compactness* hyper-parameter. By adjusting the two hyper-parameter inputs (loss

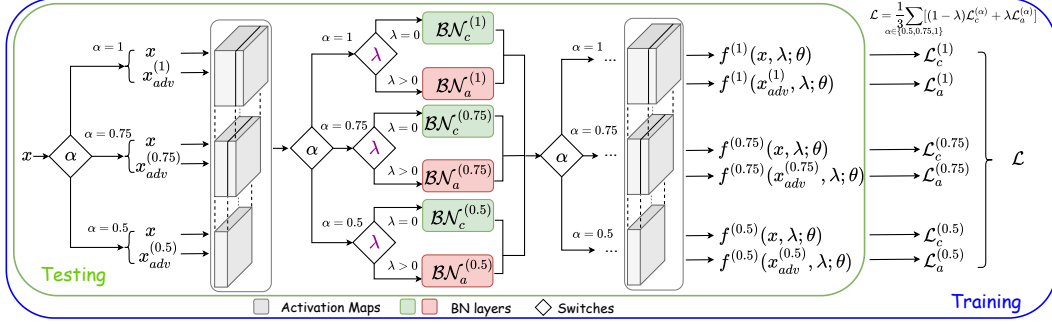

Figure 3: The OATS framework illustration, with both hyperparameters $\lambda$ and $\alpha$ (width factor) as inputs. All superscripts indicate the width factor of corresponding subnetwork. $\lambda$ controls whether to use $BN_c$ or $BN_a$. $\alpha$ controls which subnetwork to use. For example, if $\alpha = 0.5$, adversarial images, denoted as $x_{adv}^{(0.5)}$ in this case, are generated using the subnetwork with $0.5$ channel width (bottom row in the figure), and both $x$ and $x_{adv}^{(0.5)}$ are forwarded by the $0.5$ channel width subnetwork.

weight $\lambda$, and model compactness), models trained by OATS can simultaneously explore the spectrum along the accuracy, robustness and model efficiency dimensions, yielding the three's in-situ trade-off.

Inspired by the state-of-the-art in-situ channel pruning method, *slimmable network* [21], we implement the model compactness level parameter as the channel *width factor* as defined in [21]. Each width factor corresponds to a subnetwork of the full network. For example, the subnetwork with width factor $0.5$ only owns the (first) half number of channels in each layer. We could pre-define a set of high-to-low allowable widths, in order to meet from loose to stringent resource constraints. The general workflow for OATS, as shown in Fig. 3 and summarized in Algo. 2, is then similar with OAT, with the only main difference that each BN (either $BN_c$ or $BN_a$) in OAT is replaced with a switchable batch normalization (S-BN) [21], which will lead to independent BNs for different-width sub-networks. As a result, each subnetwork uses its own $BN_c$ and $BN_a$. For example, assuming three different widths being used, then every one BN in the

---

**Algorithm 2:** OATS Algorithm Outline

**Input:** Training set $\mathcal{D}$, $P_\Lambda$, model $f$, maximum steps $T$, a list of pre-defined width factors.
**Output:** Network parameter $\theta$.
**for** $t = 1$ **to** $T$ **do**
    Sample a minibatch of $(\boldsymbol{x}, \boldsymbol{y})$ from $\mathcal{D}$;
    Sample a minibatch of $\lambda$'s from $P_\Lambda$;
    Clear gradients: *optimizer.zero_grad()*;
    **for** width factor **in** width factor list **do**
        Switch the S-BN to current width factor on network $f$ and extract corresponding sub-network;
        Generate $\boldsymbol{x}_{adv}$ (using PGD);
        Compute loss in Eq. (3):
        $loss = \mathcal{L}(x, y, \lambda)$;
        Accumulate gradients: *loss.backward()*;
    **end for**
    Update $\theta$ by Eq. (3): *optimizer.step()*;
**end for**

---

original backbone will become two S-BNs in OATS, and hence a total of six BNs. To train the network, we minimize the average loss in Eq. (3) of all different-width sub-networks.

## 4 Experiments

### 4.1 Experimental setup

**Datasets and models** We evaluate our proposed method on WRN-16-8 [40] using SVHN [41] and ResNet34 [42] using CIFAR-10 [43]. Following [44], we also include the STL-10 dataset [45] which has fewer training images but higher resolution using WRN-40-2. All images are normalized to $[0, 1]$.

**Adversarial training and evaluation** We utilize $n$-step PGD attack [6] with perturbation magnitude $\epsilon$ for both adversarial training and evaluation. Following [6], we set $\epsilon = 8/255$, $n = 7$ and attack step size $2/255$ for all experiments. Other hyper-parameters (*e.g.*, learning rates, etc.) can be found in Appendix A. Evaluation results on three other attacks are shown in Appendix 4.4.

**Evaluation metrics**   **Standard Accuracy (SA)**: classification accuracy on the original clean test set. SA denotes the (default) *accuracy*. **Robust Accuracy (RA)**: classification accuracy on adversarial images generated from original test set. RA measures the *robustness* of the model. To more directly evaluate the trade-off between SA and RA, we define **SA-RA frontier**, an empirical Pareto frontier between a model's achievable accuracy and robustness, by measuring the SAs and RAs of the models dedicatedly trained by PGD-AT with different (fixed) $\lambda$ values. We could also vary $\lambda$ input in the OAT-trained models, and ideally we should hope the resulting SA-RA trade-off curve to be as close to the SA-RA frontier as possible.

**Sampling** $\lambda$   Unless otherwise stated, $\lambda$s are uniformly sampled from the set $\mathbb{S}_1 = \{0, 0.1, 0.2, 0.3, 0.4, 1\}$ during training in an element-wise manner, *i.e.*, all $\lambda$s in a training batch are *i.i.d.* sampled from $\mathbb{S}_1$. During testing, $\lambda$ could be any seen or unseen value in [0,1], depending on the standard/robust accuracy requirements, as discussed in Section 3.2.

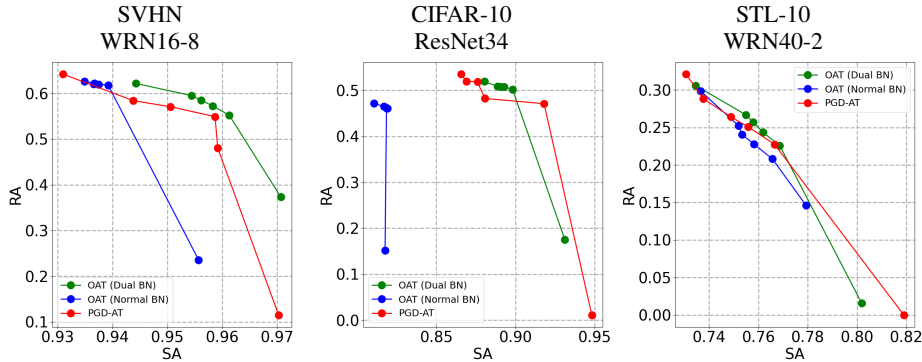

Figure 4: OAT: SA-RA Trade-off of different methods on three datasets. $\lambda$ varies from the largest value to the smallest value in $\mathbb{S}_1$ for the points from top-left to bottom-right on each curve. Results are also shown in a different form in Fig. 8 ($\lambda$-SA/RA curve) in Appendix B for readers' reference.

## 4.2   Evaluation results for OAT

Evaluation results on three datasets are shown in Fig. 4. We compare three methods: the classical PGD-AT, our OAT with normal BN, and OAT with dual BN.

**Comparison with PGD-AT (SA-RA frontier)** shown in Fig. 4, OAT (dual BN) models can achieve wide range in-situ tradeoff between accuracy and robustness, which are very close or even surpassing the SA-RA frontier (red curves in Fig. 4). For example, on SVHN dataset, a single OAT (dual BN) model can be smoothly adjusted from the most robust state with 94.43% SA and 62.16% RA, to the most accurate state with 97.07% SA and 37.32% RA, by simply varying the value of input $\lambda$ at testing time.

**Effectiveness of dual BN for OAT**   As shown in Fig. 4, OAT (dual BN) generally achieves much better SA-RA trade-off compared with OAT (normal BN), showing its necessity in model-conditional adversarial learning. One interesting observation is that, OAT (normal BN)

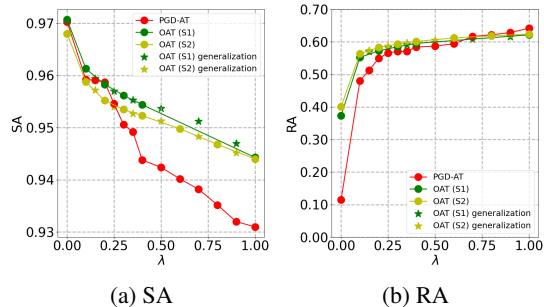

(a) SA          (b) RA

Figure 5:   Generalization to unseen $\lambda$s of OAT models. Green and yellow circles are OAT models under $\lambda$s from training sets ($\mathbb{S}_1$ or $\mathbb{S}_2$). Green and yellow stars are OAT models generalized to $\lambda$s outside training set (in $\mathbb{S}_3$). Red circles are PGD-AT models trained with fixed $\lambda$s in $\mathbb{S}_2 \cap \mathbb{S}_3$.

models can easily collapse to fit only clean or adversarial images, which is in alignment with our observations on the difference between standard and robust feature statistics shown in Fig. 2. For example, on SVHN dataset, when $\lambda$ ranges from $1.0$ to $0.1$, OAT (normal BN) collapse at almost identical RA values (ranging from 61.81% to 62.62%) with low SA (less than $94\%$) at all time (blue curve in the first column of Fig. 4). In other words, the model "collapse" to fitting adversarial images and overlooking clean images regardless of the $\lambda$ value. In contrast, OAT with dual BN can

successfully fit different SA-RA trade-offs given different $\lambda$ inputs. Similar observations are also drawn from the other two datasets.

**Sampling set of $\lambda$ in OAT training** Our default sampling set $\mathbb{S}_1 = \{0, 0.1, 0.2, 0.3, 0.4, 1.0\}$ is chosen to be sparse for training efficiency. To investigate in the influence of $\lambda$ sample set on OAT, we try another denser $\mathbb{S}_2 = \{0, 0.1, 0.2, 0.3, 0.4, 0.6, 0.8, 1.0\}$. Results on SVHN dataset are shown in Fig. 5. As we can see, OAT on $\mathbb{S}_1$ and $\mathbb{S}_2$ have almost identical performance ($\mathbb{S}_1$ has slightly better SA while $\mathbb{S}_2$ has slightly better RA), and both have similar or superior (*e.g.*, at $\lambda = 0.3, 0.4$) performance compared with fixed $\lambda$ PGD-AT. Hence, it is feasible to train OAT with more $\lambda$ samples, even sampling continuously, although the training time will inevitably grow (see Appx. C for more discussions). On the other hand, we find that training on the sparse $\mathbb{S}_1$ already yields good SA-TA performance, not only at several considered $\lambda$ values, but also at unseen $\lambda$s: see the next paragraph.

**Generalization to unseen $\lambda$** OAT samples $\lambda$ from a discrete set of values and embed them into high dimensional vector space. One interesting question is that whether OAT model have good generalization ability to unseen $\lambda$ values within the similar range. If yes, we can generalize OAT model to explore an infinite continuous set (*e.g.*, $[0, 1]$) during inference time, allowing for more fine-grained adjustable trade-off between accuracy and robustness. Experimental results on SVHN are shown in Fig. 5. The two OAT models are trained on $\mathbb{S}_1$ and $\mathbb{S}_2$ respectively, and we test their generalization abilities on $\mathbb{S}_3 = \{0.15, 0.25, 0.35, 0.5, 0.7, 0.9\}$. We also compare their generalization performance with PGD-AT models dedicatedly trained on $\mathbb{S}_3$. As we can see, OAT trained on $\mathbb{S}_1$ and OAT trained on $\mathbb{S}_2$ have similar generalization abilities on $\mathbb{S}_3$ and both achieve similar or superior performance compared with PGD-AT dedicatedly trained on $\mathbb{S}_3$, showing that sampling from the sparser $\mathbb{S}_1$ is already enough to achieve good generalization ability on unseen $\lambda$s.

More visualization and analysis could be checked from the appendices (*e.g.*, Jacobian saliency [46] in Appendix 4.5).

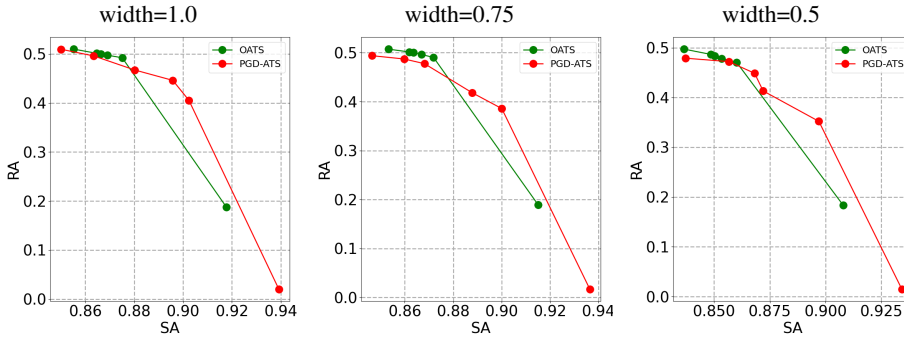

Figure 6: OATS: SA-RA Trade-off of different methods on CIFAR-10 ResNet34 with different widths. Left, middle, right columns are the full network, 0.75 width, and 0.5 width sub-network respectively. $\lambda$ varies from the largest to the smallest value in $\mathbb{S}_1$ for the points from top-left to bottom-right on each curve. The same results are also shown in a different form in Fig. 9 ($\lambda$-SA/RA curve) in Appendix B for the readers' reference.

## 4.3 Evaluation results for OATS

**Baseline method** We first design the baseline method named *PGD adversarial training and slimming* (PGD-ATS), by replacing the classification loss in original slimmable network with the adversarial training loss function in Eq. (1). Models trained by PGD-ATS have fixed accuracy-robustness trade-off but can adjust the model width "for free" during test time. In contrast, OATS has in-situ trade-off among accuracy, robustness and model complexity altogether.[2] Three width factors, 0.5, 0.75 and 1.0, are used for both PGD-ATS and OATS, as fair comparison.[3]

The results on CIFAR-10 are compared in Fig. 6.[4] As we can see, OATS achieve very close SA-RA trade-off compared with PGD-ATS, under all channel widths. For example, at width $0.75$, the most robust model achievable by OATS (top-left corner point of green curve) has even higher SA and RA ($0.67\%$ for SA and $1.34\%$ for RA) compared with the most robust model achievable by PGD-ATS (top-left corner point of red curve).

## 4.4 Evaluation on more attacks

In this section, we show that the advantage of OAT over PGD-AT baseline holds across multiple different types of adversarial attacks. More specifically, we evaluate model robustness on three new attacks: PGD-20[5], MI-FGSM [47], and FGSM [2], besides the one (PGD-7) used in the main text. For PGD-20 attack, we use the same hyper-parameters as PGD-7, except increasing $n$ from 7 to 20, resulting in a stronger attack than PGD-7. For MI-FGSM, we set perturbation level $\epsilon = 8$, iteration number $n = 10$ and decay factor $\mu = 1$ following the original paper [47]. For FGSM, we set perturbation level $\epsilon = 8$. Results on SVHN dataset are shown in Figure 7. Under all three different attacks, OAT (dual BN) models can still achieve in-situ tradeoff between accuracy and robustness over a wide range, that remains close to or even surpasses the SA-RA frontier (PGD-AT baseline).

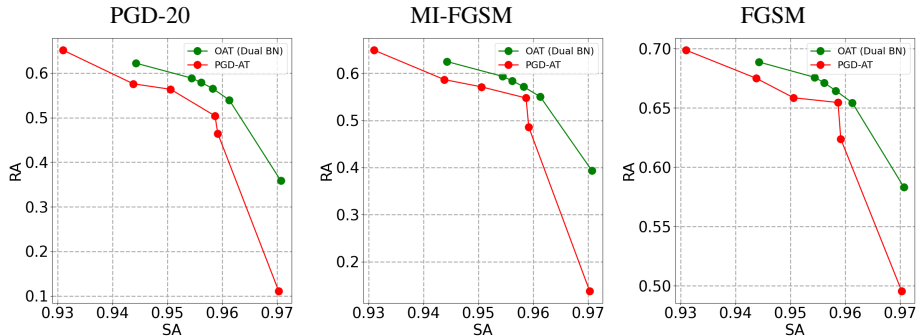

Figure 7: SA-RA trade-off on more different attacks on SVHN dataset. $\lambda$ varies from the largest to the smallest value in $\mathbb{S}_1$ for the points from top-left to bottom-right on each curve.

## 4.5 Visual interpretation by Jacobian saliency

Jacobian saliency, *i.e.*, the alignment between Jacobian matrix ($\nabla_{\boldsymbol{x}} \mathcal{L}_c$) and the input image $\boldsymbol{x}$, is a property desired by robust DNNs both empirically [8] and theoretically [46]. We visualize and compare Jacobian saliency of OAT and PGD-AT on SVHN, CIFAR10 and STL10 in Figures 10, 11 and 12 (in Appendix D), respectively. As we can see, the Jacobian matrices of OAT models align with the original images better as $\lambda$ (and also the model robustness) increases. This phenomenon provides an additional evidence that our OAT model has learned a smooth transformation from the most accurate model to the most robust model. We also find that under the same $\lambda$, OAT model has better or comparable Jacobian saliency compared with PGD-AT models.

## 5 Conclusion

This paper aims to address the new problem of achieving in-situ trade-offs between accuracy and robustness at testing time. Our proposed method, once-for-all adversarial training (OAT), is built on an innovative model-conditional adversarial training framework and applies dual batch normalization structure to pack the conflicting standard and adversarial features into one model. We further generalized OAT to OATS, achieving in-situ trade-off among accuracy, robustness and model complexity altogether. Extensive experiments show the effectiveness of our methods.

## Broader Impact

Deep neural networks are notoriously vulnerable to adversarial attacks [1]. With the growing usage of DNNs on security sensitive applications, such as self-driving [3] and bio-metrics [4], a critical concern has been raised to carefully examine the model robustness against adversarial attacks, in addition to their average accuracy on standard inputs. In this paper, we propose to tackle the new challenging problem on how to quickly calibrate a trained model in-situ, in order to examine the achievable trade-offs between its standard and robust accuracies, without (re-)training it many times. Our proposed method is motivated by the difficulties commonly met in our real-world self-driving applications: how to adjust an autonomous agent in-situ, in order to meet the standard/robust accuracy requirements varying over contexts and time. For example, we may expect an autonomous agent to perceive and behave more cautiously, (*i.e.*, to prioritize improving its robustness), when it is placed in less confident or adverse environments. Also, our method provides a novel way to efficiently traverse through the full accuracy-robustness spectrum, which would help more comprehensively and fairly compare models' behaviors under different trade-off hyper-parameters, without having to retrain. Our proposed methods can be applied to many high-stakes real world applications, such as self-driving [3], bio-metrics [4], medical image analysis [48] and computer-aided diagnosis [49].

## Footnotes

*The first two authors contributed equally.

[2]OATS models (1.18 GFLOPs) only have a tiny FLOPs overhead (around 1.7% on ResNet34), brought by FiLM layers and MLPs, compared with PGD-ATS baseline models (1.16 GFLOPs).

[3]The original paper [21] sets four widths including a smallest factor of 0.25. However, we find that while 0.25 width can still yield reasonable SA, its RA is too degraded to be meaningful, due to the overly small model capacity, as analyzed by [9]. Therefore, we only discuss the three width factors 0.5, 0.75, and 1 in our setting.

[4]Results of OATS with normal S-BN are not shown here since its performance is incomparable with other methods and the curve will be totally off range. We include those results in Fig. 9 for the readers' reference.

[5]Here we denote $n$-step PGD attack as PGD-$n$ for simplicity.

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
