[Supplementary Material]

## A   More detailed hyper-parameter settings

In this section, we provide more detailed hyper-parameter settings as a supplementary to Section 4.1. All models on SVHN, CIFAR-10, STL-10 are trained for 80, 200, 200 epochs respectively. SGD with momentum optimizer and cosine annealing [50] learning rate scheduler are used for all experiments. Momentum and weight decay parameter are fixed to $0.9$ and $5 \times 10^{-4}$ respectively. We try all learning rates in $\{0.1, 0.05, 0.01\}$ for all experiments. We report the results of the best performing hyper-parameter setting for each experiment.

## B   $\lambda$-accuracy plots

In this section, we provide a new way to present the same results shown in Figures 4 and 6, by comparing SA/RA of different methods under different $\lambda$s in Figures 8 and 9, for the readers' reference.

Figure 8: Comparison of trade-off between accuracy and robustness of different methods on three datasets. Top and bottom row show SA and RA under different $\lambda$'s respectively.

## C   More discussions on $\lambda$ sampling set

**Discrete v.s. continuous sampling**   Uniformly sampling $\lambda$ from the continuous set $[0, 1]$ achieves similar results as sampling from discrete and sparse $\mathbb{S}_1$ (within $\pm 0.2\%$ for SA/RA on SVHN), but requires $10\%$ more epochs to converge. We also empirically find sampling small lambdas more densely converges faster.

**OAT (normal BN) trained without $\lambda = 0$**   As discussed in Section 3.2, standard ($\lambda = 0$) and adversarial ($\lambda \neq 0$) features have very different BN statistics, which accounts for the failure of OAT with normal BN (when trained on both $\lambda = 0$ and $\lambda \neq 0$) and motivates our dual BN structure. One natural question to ask is: will OAT (normal BN) achieve good performance when it is trained only on $\lambda$s unequal to 0? Experimental results show that OAT (normal BN) trained without $\lambda = 0$ (*e.g.*, on

Figure 9: Comparison of OATS with baseline PGD-ATS on CIFAR-10 with ResNet34 backbone. Top and bottom row show SA and RA under different $\lambda$s respectively. Left, middle, right columns are the full network, $0.75$ width, and $0.5$ width sub-network respectively.

$\mathbb{S}_4 = \{0.1, 0.2, 0.3, 0.4, 1.0\}$) achieve similar performance with PGD-AT baselines (within $\pm 0.5\%$ SA/RA on CIFAR10) at $\lambda > 0$. But its best achievable SA ($91.5\%$ on CIFAR10) is much lower than that of OAT with dual BN ($93.1\%$ on CIFAR10).

## D  Visual interpretation by Jacobian saliency

In this section, we compare Jacobian saliency of OAT with PGD-AT, as discussed in Section 4.5. Visualization results on SVHN, CIFAR10 and STL10 are shown in Figures 10, 11 and 12, respectively.

## E  Ablation on encoding of $\lambda$

In this section, we investigate the influence of three different encoding schemes on OAT:

- No encoding (None). $\lambda$ is taken as input a scalar, *e.g.*, 0.1, 0.2, etc.
- DCT encoding (DCT-$d$). The $n$-th $\lambda$ value in $\mathbb{S}_1$ is mapped to the $n$-th column of the $d$-dimensional DCT matrix [51]. For example, 0 is mapped to the first column of the $d$-dimensional DCT matrix.
- Random orthogonal encoding (RO-$d$). Similar to DCT encoding, the $n$-th $\lambda$ value is mapped to the $n$-th column of a $d$-dimensional random orthogonal matrix.

Results of OAT with different encoding schemes on CIFAR-10 are shown in Figure 13. As we can see, using encoding generally achieves better SA and RA compared with no encoding. For example, the best SA achievable using RO-16 and RO-128 encoding are $93.16\%$ and $93.68\%$ respectively, which are both much higher than the no encoding counterpart at $92.53\%$. We empirically find RO-128 encoding achieves good performance and use it as the default encoding scheme in all our experiments.

(a) Original images in SVHN test set

(b) Jacobian saliency maps of OAT models

(c) Jacobian saliency maps of PGD-AT models

Figure 10: Jacobian saliency maps of OAT and PGD-AT models on SVHN. For (b) and (c), in each column are saliency maps of corresponding images in the same column of (a); in each row are saliency maps of models under different $\lambda$s ($\lambda = 0, 0.1, 0.2, 0.3, 0.4, 1.0$ from top row to bottom row).

(a) Original images in CIFAR-10 test set

(b) Jacobian saliency maps of OAT models

(c) Jacobian saliency maps of PGD-AT models

Figure 11: Jacobian saliency maps of OAT and PGD-AT models on CIFAR-10. For (b) and (c), in each column are saliency maps of corresponding images in the same column of (a); in each row are saliency maps of models under different $\lambda$s ($\lambda = 0, 0.1, 0.2, 0.3, 0.4, 1.0$ from top row to bottom row).

(a) Original images in STL-10 test set

(b) Jacobian saliency maps of OAT models

(c) Jacobian saliency maps of PGD-AT models

Figure 12: Jacobian saliency maps of OAT and PGD-AT models on STL-10. For (b) and (c), in each column are saliency maps of corresponding images in the same column of (a); in each row are saliency maps of models under different $\lambda$s ($\lambda = 0, 0.1, 0.2, 0.3, 0.4, 1.0$ from top row to bottom row).

(a) SA

(b) RA

Figure 13: Results of OAT with different encoding schemes on CIFAR-10.