[Reviews · NeurIPS 2020]

Review 1

Summary and Contributions: This paper aims to explore the balance between robustness and accuracy, by proposing training methods to work on regularisation terms including both classification loss and robustness loss. It considers an approach that can make the training algorithm conditioned on lambda -- a hyper-parameter.

Strengths: A framework for adversarial training that can balance multiple regularisation terms.

Weaknesses: Based on the idea of conditioning, some detailed study is conducted, although the key technical methods are all from other papers. For example, it uses Feature-wise Linear Modulation from [31] for the initial implementation, but discovers that the performance is significantly degraded. Then, a dual batch normalization (dual BN) from [16] is considered. Afterwards, the framework is extended to work with the other dimension on efficiency. Technically, my concern is with the seemingness small number of sampled lambda values and that the lambda values are sampled uniformly. Is there any theoretical argument that such sampling can actually be able to give us a sufficiently good picture on what is going on in terms of the tuning of hyper-parameter. More specifically, it is unclear to me how to validate the resulting framework beyond the simple comparisons as e.g., in Fig 5 and Fig 6. At least I am not convinced that the balance between SA and RA is so simple. It might be useful to also discuss the cost of this framework. From Fig 3, it looks to me it is training a network of multiple times larger than the original network. Based on the above observation, I think the problem itself is important and the general approach is interesting, with my conservations on the novelty of this work -- given that it is following the conditional training approach and adapted from other existing methods ([31],[16], etc), the novelty is not as significant. Also, it is unclear to me how good this approach can be used to study the robustness-acccuracy balance -- more theoretical argument might be needed.

Correctness: the method looks interesting to me, although I am a bit conservative on if it is able to solve, or is the ultimate tool to analyse, the robustness-accuracy balance.

Clarity: it is well written

Relation to Prior Work: related work is discussed, and cited

Reproducibility: Yes

Additional Feedback:


Review 2

Summary and Contributions: This paper describes a novel “in-situ” flexible model that can trade-off between accuracy and robustness (as well as compactness, as an extension), at test time based on user specification. For the first time such problem is raised, formulated and addressed in the adversarial defense literature.

Strengths: The method was invented to tackle an important and under-addressed problem: trading off accuracy and robustness using adversarial training, without re-training. The proposed solution is a plug-n-play extension to standard AT, based on “model-data space joint sampling” (as called by authors). Another interesting extension was also considered for co-optimizing the network width. I think the overall methodology is sound and novel. Figure 2 visualization of BN features explains the challenge of packing different SA-RA models into one. Supplement D proves that dual BN is indeed a knob of method success. Experimental results show that CAT (dual BN) works well in general, being comparable to traditional AT training for each individual lambda parameter. The authors also presented a large number of ablation experiments, making their conclusion more convincing.

Weaknesses: IMHO, the SA-RA trade-off seems to be most useful when human is placed in the loop to decide “how aggressive” the predictor shall behave as. For example, besides standard image classification, it would be meaningful to benchmark the proposed approach on medical image classification, or other human-in-loop decision making, for showing its real benefits. It seems lambda is specified by the user during test time. But I am not sure for users, what is the principle or rule of thumb here, for selecting the lambda value to achieve their desired performance. There is still room for performance improvement. It shows that retraining can still boost TA-RA in some data points. Also, CATS is (understandably) a bit outperformed by PGD-AT, hence jointly optimizing three objectives seem to be challenging. There is no discussion of training costs. Does CAT take longer to train compared to PGD-AT?

Correctness: Yes

Clarity: Yes

Relation to Prior Work: Yes

Reproducibility: Yes

Additional Feedback:


Review 3

Summary and Contributions: The paper proposed an empirical framework called calibratable adversarial training. It allows for user-specified calibration of desired robustness level, depending on test-time use case, without re-training. Their experiments show that, with just “once for all” training, this method impressively achieves similar or superior performance, compared to dedicatedly trained models at various configurations.

Strengths: This is the first framework proposed for a new goal: to control the trade-off between accuracy and adversarial accuracy “in-situ”. The motivation is clear and interesting: their new goal is meaningful for quickly exploring the accuracy-robustness performance spectrum of a model, avoiding repetitive training to exhaust hyperparameters. It also enables future applications where the perceiving agent may switch between “optimistic” and “conservative” from time to time, as reacting to the dynamic environment. The framework is enabled by taking the trade-off parameter as a conditional input and is trained with sampling this additional parameter differently at each minibatch (the loss is adapted accordingly). This is a sound idea, and the authors also discussed the impact of sampling strategy in their experiments. The idea is extended to incorporate another parameter controlling model complexity. Their methodology hence appears to be general. Experiments compared with strong PGD-AT baselines. The authors describe their protocols in detail, and the results look convincing (comparable to adversarial training but avoiding re-training). I like the authors plotting the SA-RA frontiers that nicely summarize the big picture of algorithm performance spectrum. From the supplementary, the Jacobian visualizations, and the demonstrated robustness to more unseen attack types, are also noted.

Weaknesses: This is a strong paper that I feel overall positive about. No particular weakness is required for authors to address during rebuttal; just some nitpicks and suggestions: - I am quite curious what will happen, if you feed in lambda outside [0,1] range, at test time? - For the role of dual BN in adversarial robustness, a better reference than [16] is “Intriguing properties of adversarial training at scale”, ICLR 2020 (from the same authors). - As one key building block, the introduction of FiLM layer is underwhelming, and more details should be included to make the paper self-contained. - As an experimental paper, it would be nicer to demonstrate results on larger datasets. This is just a suggestion afterwards; I understand adversarial training on ImageNet scale cannot be completed easily in short time. - (minor) You might want to discuss a recent paper of similar taste, “Once-for-All: Train One Network and Specialize it for Efficient Deployment on Diverse Hardware Platforms”, although admittedly the two works solve different questions.

Correctness: Yes

Clarity: This paper is well-written, and the logic flow is easy to follow. Contributions are laid out clearly. Figures 1 to 3 are nicely done and helpful for understand the algorithms.

Relation to Prior Work: Yes

Reproducibility: Yes

Additional Feedback:


Review 4

Summary and Contributions: This paper studies an important problem in adversarial machine learning: how to flexibly switch between different levels of robustness and accuracy tradeoff without retraining the models. Motivated by an existing work that applies batch normalization separately for clean and adversarial data, this paper proposes a calibratable adversarial training (CAT) scheme to achieve different levels of in-situ robustness-accuracy tradeoff all at once in a single round of training. At test time, this allows the user to obtain any level of robustness/accuracy by specifying a tradeoff parameter. A slimming parameter is further incorporated into the CAT framework for runtime efficiency. The proposed methods are verified on three datasets, along with an explanation for the clean-adversarial batch norm separation. ----------- I am happy to raise my score to 6 after the rebuttal. Most of my concerns have been addressed to some extent. While I still think the unified formulation is not strictly precise, I appreciate the authors' attempts. Overall, the idea proposed in this paper is very novel.

Strengths: 1. This paper is very well written and easy to read. 2. The proposed CAT/CATS training schemes are well-motivated, and could benefit a wide range of industrial applications. 3. The idea of batch norm separation can be easily implemented, and seem to work well with adversarial training. 4. Complete experimental evaluation and explanations.

Weaknesses: 1. Novelty of the clean vs adversarial statistics separation via isolated batch normalization is limited, considering a similar technique has been proposed in [16]. Although [16] is not for adversarial training, but technically, it proves that data statistics can be separated without harming the learning (it even improves the learning). 2. The proposed CAT/CATS training is very similar to ensembling, but with shared weights for Conv layers, which is also similar to the weight sharing across child networks in neural architecture search (NAS). I do acknowledge the contribution of transferring these ideas to adversarial training. 3. The demonstrated robustness-accuracy-runtime tradeoffs in Figures 4 and 6 are not even over different parameters. RA doesn’t change much when SA is small, but suddenly drops to a much lower level when SA is above some values. This means the proposed methods did not really address the trade-off issue, but simply put together different subnetworks. 4. The formulation of existing adversarial training methods in Equation (1) and (2) are wrong! In Eq. (1), left is not equal to right: left is a minimization problem, right is just an empirical error, missing min_{\theta} on the right. The formulation of PGD-AT in Equation (2) is wrong, L_c=0 for PGD-AT, and the hyper-parameter is not (1-lambda)/(lambda) for TRADES (not sum up to one). These formulations have been well summarized in one ICLR20 paper [2]. Please double check the original formulation used in [6], by its Equation (2.1). There is also no L-c term in MMA, see its Equation (3). 5. A followed up issue, I am not sure what it is for PGD-AT with \lambda != 1. For example in Figure 5, the authors tested different \lambda for PGD-AT, where it is hard to interpret what that means for PGD-AT (it is no longer the PGD-AT if \lambda != 1). 6. The evaluation metric RA uses the same PGD attacks for both training and testing, which is a bit too weak, should use a stronger PGD attack for testing (for example, train on PGD-10, test on PGD-40). This may not change the conclusion of this paper though. And also, the experiments in Figure 5 should be done on CIFAR-10 not SVHN, otherwise it is hard to interpret the real difference compared to standard PGD-AT, as most of the understanding of PGD-AT in this field is on CIFAR-10.

Correctness: Claims and method are correct, though there are formulation issues.

Clarity: Yes, very well written.

Relation to Prior Work: Yes.

Reproducibility: Yes

Additional Feedback:

[Author Response · NeurIPS 2020]

## General questions

▷ **Lack of novelty (R1, R5)** We humbly disagree that our proposed method lacks novelty. As noted by both R2 and R3, our paper raised, formulated and addressed a novel problem to achieve an "in-situ" flexible model that can trade-off between accuracy and robustness at test time based on user specification, for the first time in the adversarial defense literature. Our proposed framework is featured by a joint model-data sampling method, which takes the trade-off parameter $\lambda$ as a conditional input and trains the network with sampling this additional parameter differently at each minibatch (the loss is adapted accordingly). The above problem definition and "joint model-data sampling method" are the main novelties of the paper, as acknowledged by both R2 and R3. Although we leverage dual BN and FiLM as the off-the-shelf tools (we never claimed them as our novelty), their usages are also properly customized to fit our framework. For dual BN, we observe that the conflict between standard and adversarial feature statistics is a roadblock for our new problem of packing them in one model, and dual BN [16] can be an effective solution. Difference between our modified and the original version in [16] are summarized in lines 206-217. For FiLM, we condition on $\lambda$ while the original condition on input data.

▷ **Training cost (R1, R2)** As mentioned in footnote 1 (page 8), CATS models only have a tiny FLOPs overhead (around 1.7% on ResNet34) compared with PGD-ATS baseline models. CAT requires almost the same number of epochs to converge compared with baseline PGD-AT, and thus almost identical training time. (To R1) The model in Fig 3 is actually not large, since the dual BN structure only introduces a tiny overhead (brought by an auxiliary BN), and the S-BN enables weight sharing over channels with different widths. We will make this clearer in revision.

## Response to R1

▷ **Sampling strategy of $\lambda$.** For both the dedicated training baseline (PGD-AT) and our method, increasing $\lambda$ will increase RA and sacrifice SA, no matter how densely $\lambda$ is sampled. We empirically find our method robust to various $\lambda$ sampling strategies. Increasing $\lambda$ sampling set size (see lines 302-310) or uniformly sampling from the continuous set $[0, 1]$ (see lines 520-523 in Appx D) achieve very similar performance with our default sampling strategy from $\mathbb{S}_1$. For non-uniform sampling, if we sample from $\{0, 0.1, 0.2, 0.3, 0.4, 1\}$ with probabilities $\{0.3, 0.2, 0.2, 0.1, 0.1, 0.1\}$ respectively, the performance is still similar with our default strategy: within $\pm 0.5\%$ for SA/RA on CIFAR10.

▷ **Utility to explore trade-off.** We do not intend to claim CAT as any "ultimate" tool to solve or analyze the trade-off. One application of CAT is to sketch (approximately estimate) the empirical achievable trade-off between accuracy and robustness of the same model trained under different $\lambda$ values, in a cheap and efficient way (avoiding training many times). From our experimental results, the SA-RA trade-off curves produced by CAT are highly aligned with dedicatedly trained ones. We will make this clearer in final version. Theory for CAT will be our immediate next work.

## Response to R2 & R3

Thanks for appreciating our work. Your suggestions on writings, related works, and more experiments (including ImageNet, medical image classification, or other human-in-loop decision making systems) are highly valuable and will be addressed in the final version. (R2) We will further improve the performance in our future work. (R2) In general, users set $\lambda = 0$ to maximize standard accuracy, and increase $\lambda$ towards 1 when more robustness is demanded at some price of accuracy. In practical applications, one "rule of thumb" suggestion would be to quickly examine SA and RA at a few anchor $\lambda$s such as 0, 0.5, 1, and then test further in one most desired interval. (R3) When fed with $\lambda < 0$ or $\lambda > 1$ at test time, the performance of CAT gradually decays as expected, since it is designed to fit $\lambda$s within range $[0, 1]$ only.

## Response to R5

▷ **Relation with ensembling/NAS?** We agree that there exist relationships, and will discuss in the final version.

▷ **Interpretation of Fig 4, 6.** We apologize for the confusion. The trade-offs in Fig 4 and 6 are indeed over different parameters: $\lambda$ varies from the largest to the smallest value in $\mathbb{S}_1$, for points from top-left to bottom-right on each curve, as noted in the figure captions. Figs 4,6 show that the performance of CAT aligns with the SOTA dedicated adversarial training, that can be empirically considered as the model's best achievable trade-off. Also, our method is verified to generalize to **unseen** $\lambda$ values (see Fig 5), so it is evidently more than "simply putting together different subnetworks".

▷ **Formulation and term issues.** We will update Eq (1) as you suggested. For PGD-AT and MMA, we can include clean images in adversarial training as a standard variant (e.g., Appx B in [7], Eq. (8) in [22]), so that $\lambda \neq 1$ and $\mathcal{L}_c \neq 0$. For TRADES, we can scale the trade-off parameters to achieve sum-one, whose optimization remained unchanged. We will make our notations stricter in revision, meanwhile none of them affected our method or claims.

▷ **Evaluation on different attacks; Fig 5 on CIFAR10.** We evaluated our models defended with PGD-7 on three other different attacks (including a stronger PGD-20) in Appx B (on SVHN). We also provide RAs against PGD-40 on CIFAR10 per your suggestion here: ($\lambda$ from 0 to 1 in $\mathbb{S}_1$) **Baseline (PGD-AT)**: 1.04%, 46.5%, 47.9%, 51.1%, 51.3%, 53.2%. **Ours (CAT)**: 17.3%, 49.2%, 50.1%, 50.42%, 50.46%, 51.8%. Results in Fig 5 can also generalize to CIFAR10: ($\lambda$ from 0 to 1 in $\mathbb{S}_1 \cup \mathbb{S}_3$) **Baseline (PGD-AT)**: (SA) 94.83%, 91.81%, 89.47%, 88.17%, 88.07%, 87.9%, 87.81%, 86.90%, 86.88%, 86.68%, 86.58%, 86.57%; (RA) 1.12%, 47.1%, 47.41%, 48.25%, 48.9%, 50.61%, 51.83%, 51.99%, 53.17%, 53.35%, 53.43%, 53.52%. **Ours (CAT)**: (SA) 93.12%, 89.82%, 89.57%, 89.28%, 89.15%, 89.07%, 88.98%, 88.88%, 88.53%, 88.32%, 88.28%, 88.04%; (RA) 17.54%, 50.21%, 50.33%, 50.76%, 50.77%, 50.82%, 50.85%, 50.86%, 51.05%, 51.64%, 51.89%, 51.91%.

[Meta-Review · NeurIPS 2020]

This paper focuses on an empirical framework called calibratable adversarial training. The proposal allows for user-specified calibration of desired robustness level, depending on test-time use case, without re-training. The philosophy behind sounds quite interesting to me, namely, "Once-for-All" training. This philosophy leads to a novel algorithm design I have never seen, i.e., CAT and CATS. The clarity and novelty are clearly above the bar of NeurIPS. While the reviewers had some concerns on the significance, the authors did a particularly good job in their rebuttal. Thus, all of us have agreed to accept this paper for publication! Please carefully address R5' comments in the final version. Namely, the unified formulation should be strictly precise.